# *Roda* and *Terreiro*: The Historiography of Brazil's Visual Arts at the Crossroads of Globalization

Roberto Conduru 

Department of Art History, Southern Methodist University, Dallas, TX 75275, USA; rconduru@smu.edu

**Abstract:** The article focuses on Brazil's visual arts historiography from the 1990s onwards when institutions in Europe and the U.S. began to present Brazil's art more frequently amid the growing globalization of the art system. Edge cases are highlighted to demonstrate how scholars based outside Brazil are helping to build a canon of that country's visual arts that contrasts and surpasses the canon of Brazil's visual arts outlined in Brazil's collections, exhibitions, publications, and scholarly production. The image of *roda* (circle) in Ronald Duarte's *Nimbo/Oxalá* and Ricardo Basbaum's image/idea of "*terreiro de encontros*" (terrace of encounters) are proposed as Afro-Brazilian references with which to face the challenges of these historiographic crossroads.

**Keywords:** historiography of Brazil's visual arts; Brazil's visual arts; Afro-Brazilian art; art historiography



## 1. Opening a Circle

Ronald Duarte's (Barra Mansa, 1963) *Nimbo/Oxalá* (Nimbus/Obatala) (Figure 1), a collaborative and temporary intervention held for the first time in 2004, is both an artwork and an ex-voto for Obatala, the elder Yoruba deity, in gratitude for his wife's health recovery. Its title links the physical–chemical event with the orisha, connecting the brief and uncontrollable cloud generated by releasing a total load of fire extinguishers to the creator of human beings for the Yoruba people. Some of Obatala's attributes confirm the connection: the day of the performance, Friday; the color of the smoke, also predominant in people's clothing, white; and the temporarily formed element, cloud, and one of its qualities, diffuse omnipresence. In addition to promoting a collective greeting of thanks to Obatala, the artist explores the semantic multiplicity of Afro-Brazilian religions and their practitioners' habit of surreptitiously disseminating their signs in Brazilian cultural codes, facing persistent restrictions and attacks on sociocultural manifestations of African origin in Brazil.

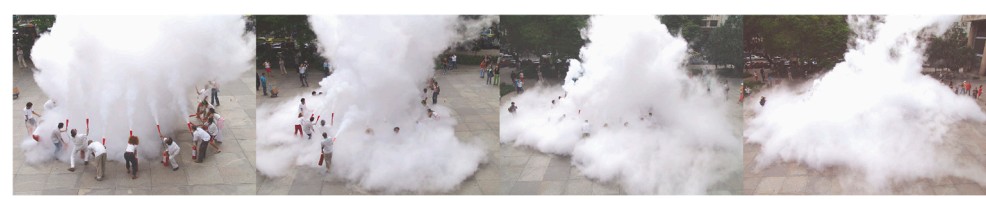

**Figure 1.** Ronald Duarte, *Nimbo/Oxalá* (Nimbus/Obatala), performance, Rio de Janeiro, 2004. Photos courtesy of Ronald Duarte.

*Nimbo/Oxalá*'s circular structure reviews the *xirê*, the ritualistic circle of Afro-Brazilian religious ceremonies in which the deities are invoked and from which they manifest through religiously initiated people in trances.[1] Just as African religious systems were reinvented in Brazil, from the forced transatlantic migration of enslaved men and women between the sixteenth and nineteenth centuries, musical rhythms derived from Africa were created in the country. One of these rhythms, *samba*, is also practiced around explicit or allusive informal circles, the "*rodas de samba*" (samba circles), in which *sambistas* (*samba* composers)

present new songs and play classic ones, (re)evaluating them while bringing together a few to many people in circles or around tables to play music, sing, dance, drink, eat, tell stories, have fun, and socialize.[2]

In Portuguese, the expression "*abrir a roda*" (to open the circle) means starting the *samba* session or expanding the ritualistic circle so that more people can enter and participate in it. Starting this article with *Nimbo/Oxalá* and concluding it with the image/idea of "*terreiro de encontros*" (terrace of encounters) by the artist Ricardo Basbaum (São Paulo, 1961), I re-open a debate about the recent production of Brazil's visual arts historiography[3]. Trained and active as an art historian in Brazil since the 1980s and working in the United States since 2018, I have lived—as an active participant, more than an observer—through the process in which, since the 1990s, agents and institutions have been substantially altering the insertion of the arts of Brazil in the world art system. By using a theoretical methodology, delineating a brief panorama of the recent historiographic production on Brazil's visual arts, especially those produced outside that country, dialoguing with authors who previously focused on this topic, and analyzing artworks, exhibitions, institutions, and, above all, crucial passages from some publications (focusing mainly on academic books), this article makes this discussion more explicit, highlights some hot spots and calls other authors to engage in the debate about this historiography, its authors, and contexts and modes of production and dissemination. The challenges, possibilities, and problems in focus here are not restricted to the Brazilian case. It is possible to observe similar historiographical tensions in the framing processes of the visual arts in Latin America and other contexts considered peripheral by the dominant institutions in the global art system. Both *roda* and *terreiro* are metaphors for a more dialogic process in writing the history of Brazil's and Latin America's visual arts.

## 2. Changes in the Historiographical Practice

Initially, it is necessary to indicate the historiography that has been constituted amid Brazil's visual arts insertion in the globalized artistic system from the 1990s onwards. Based on the centrality of the African and African diasporic cultural experience for Brazil and the modern culture, I propose *Nimbo/Oxalá* as a metaphor for art history's globalization process. One can think of art history's ideas and achievements, theories, methods, and narrative models as a dissipating cloud within Western culture's dissemination throughout the world. Thus, one can predict that it can be fully sprinkled one day, traversing all artistic and cultural institutional contexts. One question is whether, after spreading out, its unity will be kept and will remain the same discipline, still being called Art History, or whether it will become another discipline, multiple disciplines, or will disappear. Another issue is whether the globalized history of art will be the worldwide extension of traditional art history, which will improve itself to be more comprehensive but maybe also more dominant, or whether a critical historiographical practice will effectively transform itself, abandoning the hierarchical, centralizing, and colonialist practices that have characterized the discipline, to be faithful to old and new ideals.

It is better to follow Jill Casid and Aruna D'Souza's proposal "to move away from unifying or 'global' art history projects by acknowledging that 'confronting the challenge of developing practices of and for 'the global' necessarily involves learning how to engage with a range of irresolvable frictions, disunities, and incommensurabilities.'" (D'Souza 2012, p. 176). We can also assume the partial and ever-changing mosaic of incomplete histories, the tense sum and even chaos of multiple accounts of art with varied structural principles, spatial focuses, and temporal arrangements, in tune with the "plurality of trans-regional narratives," as proposed by Piotr Piotrowski (Piotrowski 2009).

Therefore, what interests me in this essay are the tensions inherent to the historiographical cloud recently configured from Brazil's arts. Names such as French Germain Bazin (Bazin 1963), English Guy Brett (Brett 2005), North Americans Robert Farris Thompson (Farris Thompson 1983) and Robert Smith (Smith 2012a, 2012b), Greek Stamo Papadaki (Papadaki 1950), and Portuguese Santos Simões (Santos Simões 1965), among others, indicate that foreign criticism and historiography of Brazil's arts are not recent. But this

historiographical process has changed drastically since the late 1980s, growing and deepening continuously as the European and North American art systems have increasingly assimilated and spread Brazil's visual arts worldwide.[4]

Although exhibitions of Brazil's visual arts had been presented outside Brazil before, the process here in focus intensified in the 1990s, outlining another art criticism and historiography with publications such as the catalogs of the first international shows of Hélio Oiticica (Oiticica 1992) and Lygia Clark (Borja-Villel and Mayo 1997). A decisive moment in this process was the acquisition of the Adolpho Leirner Collection of Brazil's constructive art by the Museum of Fine Arts, Houston (MFA Houston), in 2007, its first exhibition and corresponding catalog (Olea and Ramírez 2009). Recent examples of how this process continues are the catalogs of the exhibitions *Lygia Pape: A Multitude of Forms*, presented at the Metropolitan Museum of Art in New York in 2017 (Candela 2017), and *Tarsila do Amaral: Inventing Modern Art in Brazil*, presented by the Art Institute of Chicago and the Museum of Modern Art in New York (MoMA) in 2017 and 2018 (D'Alessandro and Pérez-Oramas 2017).

Articles in peer-reviewed journals have also been essential in configuring the outside historiography on Brazil's visual arts. It is even challenging to point out examples among many contributions. Still, with Michael Asbury's 2000 "An Experimental Exercise of Liberty" and Kaira M. Cabañas' 2022 "Art's Histories without Art History" (Asbury 2000; Cabañas 2022), one can trace a continually increasing process.

For nearly a decade and a half, academic books published in English and outside Brazil have helped shape the art historiography of Brazil's visual arts in the northern hemisphere. Scholars in the U.S. have been publishing academic books explicitly focused on Brazil's visual arts, from Claudia Calirman's 2012 *Brazilian Art under Dictatorship: Antonio Manuel, Artur Barrio, and Cildo Meireles* to Mariola V. Alvarez's 2023 *The Affinity of Neo-concretism: Interdisciplinary Collaborations in Brazilian Modernism, 1954–1964* (Calirman 2012; Alvarez 2023). If we consider academic books that include art related to Brazil and other contexts, the time scope is extended, encompassing Esther Gabara's 2008 *Errant Modernism: The Ethos of Photography in Mexico and Brazil* to her 2023 *Non-literary Fiction. Art of the Americas under Neoliberalism* (Gabara 2008, 2023). Regarding edited volumes, we can highlight *Theories of the Nonobject: Argentina, Brazil, Venezuela, 1944–1969*, edited by Mónica Amor in 2016, and *Purity is a Myth: The Materiality of Concrete Art from Argentina, Brazil, and Uruguay*, edited by Zanna Gilbert, Pia Gottschaller, Tom Learner, and Andrew Perchuk in 2021 (Amor 2016; Gilbert et al. 2021). Moreover, books are coming out soon or in a while: Matthew F. Rarey's *Insignificant Things. Amulets and the Art of Survival in the Early Black Atlantic* (Rarey 2023), and Irene Small's forthcoming book that takes Lygia Clark's notion of the "organic line" as its point of departure, tracking its emergence and comprehending it "as a generative conceptual tool, one that does expansive aesthetic, epistemological, and political work well beyond Clark's immediate context,"[5] among other promising research results.

Notably, these new generations of scholars outside Brazil have been producing an innovative scholarly record from original insights framed by current theory and based on primary resource research that has been helping to change the knowledge of Brazil's visual arts history. It is exciting how their research sometimes brings to new light artists, artworks, institutions, and topics overlooked in the art historiography produced in Brazil. I think of, for example, Isobel Whitelegg rethinking a little-discussed period of the São Paulo Biennial during the Brazilian civil–military dictatorship (Whitelegg 2009); Amy Buono analyzing books (texts and illustrations), canvas paintings, and majolica jars to argue "the ways that inventories and catalogues become sources for colonial scholarship in general and art history in particular" (Buono 2014); Mariola V. Alvarez re-examining the category of national art from the work of Manabu Mabe (Alvarez 2016); Elena Shtromberg using central systems of communication (newspapers and television), exchange (currency), and representation (maps) to analyze 1970s Brazil's visual arts (Shtromberg 2016); Kaira M. Cabañas critically analyzing the singular creation of art by mentally ill people in Brazil during modernism from dialogues and partnerships between psychiatrists, art

critics, and artists in exchanges between the medical and artistic realms (Cabañas 2018); Abigail Lapin Dardashti highlighting the Black activism of Januário Garcia's photography (Lapin Dardashti 2020); or Lucy Steeds discussing relationships between ecology, art, and historiography based on interventions by Juraci Dórea in the backlands of Bahia, in northeastern Brazil, and in art biennials in that country and abroad (Steeds 2023).

### 3. Tensions in the Historiographical Field

However, these changes in Brazil's visual arts insertion in the global artistic system have been tensioning the historiographical field. It is worth revisiting Rodrigo Naves' 2002 analysis of the problematic effects of the then-recent foreign assimilation of Brazil's visual arts and its reverberation in that country. For him, "Brazilian modern art began to be evaluated according to the flow and reflux of the dominant tendencies in a certain moment in the great cultural centers" (Naves 2002, p. 10). Still, in his view, "subjecting Brazilian art to parameters unconnected with its formation—as in the case of contemporary readings that opposes itself to the modern production—will inevitably lead to an impoverishment and simplification of what we have best in our mismatches in relation to the great centers: a complexity that is not born of a rich constitution but of a complicated historicity." He also does not appreciate how the privilege given to the relation between art and life in the works of Clark and Oiticica was internally incorporated: "the colony gladly accepted the metropolis' judgment on some of its children favored by luck" (Naves 2002, p. 18).

Six years later, in a lecture presented at ARCO'08, the International Contemporary Art Fair in Madrid, in which Brazil was the guest country, Laymert Garcia dos Santos also analyzed how the Brazilian art system lost "the ability to enforce its own criteria for contemporary production within its context" (Garcia dos Santos 2008). Articulating the acquisition of the Adolpho Leirner Collection by the MFA Houston to the same museum's project, "*Recovering the Critical Sources of Latin American/Latino Art*," he analyzed how the power to evaluate Brazil's visual arts was being transferred to the United States in that process, highlighting how "the role of the so-called Brazilian art system has been to function, at worst, as a spectator, at best, as a kind of adjunct in a game in which we enter with works, with expertise and even with financial resources to promote elsewhere the development of a new niche explored by the so-called 'creative industries'" (Garcia dos Santos 2008).

Unfortunately, the process criticized by Naves and Garcia dos Santos is moving forward at full speed. In successive visits to MoMA, I could observe how art from non-European and non-U.S. contexts has been incorporated into its long-duration exhibition of art since modernism. In March 2014, I experienced a few works made by some Brazilian artists who participated in the Neo-Concrete Movement as a complimentary part of MoMA's main exhibition, more precisely in room 26, on the fourth floor, an auxiliary enclosure with a staircase connecting to the fifth floor and somewhat aside the main narrative flow that articulated almost exclusively European and North American achievements. A year later, nothing from Brazil was presented in that peripherical room or the long-term exhibition, which was very similar to the configuration it had a year earlier. Both change and continuity indicate how room 26 was used for short-term complementary exhibitions and, therefore, secondary in MoMA's history of modern art. The situation significantly changed after the more recent expansion of that museum, which reopened to the public in October 2019. In a 2020 visit, I could experience more works by Brazilian artists from different periods scattered in MoMA's master narrative about modernism; what had been temporarily presented on the sidelines not only increased in number but also became less transient and was integrated into the long-term exhibition. Even though these additions blurred its focus and softened its strength, they did not structurally affect its main tenets and central narrative. What Ana Letícia Fialho stated in 2005 remains valid: "The museum still hasn't reinterpreted the history of art in a way that embodies the unique contributions of Latin-American" (Fialho 2005b) and other so-called peripheral artists' works.

But my question here is less about which artists, artistic movements, and artworks were included or should belong to the world canon of art since modernism and more about

how, where, and by whom the history of Brazil's visual arts has been and will be written and conducted. Since institutions in Europe and the U.S. began to present Brazil's art more frequently, from the 1990s onwards, amid the growing globalization of the art system, curators, collectors, and scholars based outside Brazil have been building a particular canon of Brazil's visual arts. They have been establishing a canon that differs from the one outlined in Brazil's collections, exhibitions, publications, and scholarly production in different types of publications.

Since Fialho's, Garcia dos Santos,' and Naves' critical appraisals, this process has stepped up and improved without losing some negative features. As expected, in an unequal globalized conjuncture and in a somewhat parochial cultural context eager to consume its image forged by agents located in the dominant artistic centers, the external canon of Brazil's visual arts has already reverberated there, was assimilated in various ways by different agents of the art system, and generated critical reflections on this process.

Indeed, art historiography does not have ideal, naturally determined, or even privileged agents or places of production. The debate on Brazil's visual arts needs to be further expanded based on contributions from agents situated in other contexts, whether historically linked or not to Brazil. Still, art historiography is not immune to institutional or individual subjectivism and market interests. Just as nationalism crosses borders, capitalist coloniality permeates the global process of modernization (Quijano 2000), constituting interconnected cultural systems that replicate centers and peripheries as instances of power, affecting the historiography of art in Brazil and beyond. Hence, there is no surprise in the exchanges of dominant agents, institutions, and centers in the global art system, which explains, for example, the centrality of southeastern Brazil's art in historiography written in Brazil and abroad, and how art history has been presented, written, and taught from articulations between arts philanthropic funding institutions in the U.S. and Brazilian artistic, teaching, and researching institutions.

Perhaps somewhat naively, one could argue that the history of Brazil's visual arts produced outside the country could help the art historiography written there to escape the parochial nationalism that almost always characterized it, just as a more effective dialogue with the scholarly record published in Brazil could help mitigate the imperialism intrinsic to the North Atlantic art historiography. The 2012 *Third Text* special issue "'Bursting on the Scene:' Looking Back at Brazilian Art" edited by Sergio Bruno Martins (Martins 2012), the 2017 *Ars* special issue on Hélio Oiticica edited by Dária Jaremtchuk (Jaremtchuk 2017), and the 2021 *Brésil(s)* dossier "*Le Populaire et le Moderne: l'Art Brésilien, 1950–1980*" (The popular and the modern: Brazilian art, 1950–1980) edited by Abigail Lapin Dardashti and Ana Magalhães (Lapin Dardashti and Magalhães 2021) bring together specialists from different generations, ethnic backgrounds, and sociocultural contexts to advance the field.

However, the historiographic tension highlighted by Naves, Fialho, and Garcia dos Santos only intensifies as interest in Brazil's visual arts outside the country increases. The 2016 Colección Patricia Phelps de Cisneros's gift to MoMA, which added more than 100 modern artworks from Latin America into its collection, encourages comparisons with MFA Houston's incorporation of the Adolpho Leirner Collection. Produced at a time when there were very few specialists in Brazil's visual arts outside that country, the MFA Houston 2009 exhibition catalog of the Adolpho Leiner Collection includes contributions from authors born and based in different American countries (Olea and Ramírez 2009). The catalog of the exhibition *Sur Moderno: Journeys of Abstraction—The Patricia Phelps de Cisneros Gift*, presented at MoMA in 2019–2020, is an example of how the privilege granted to a bibliography produced in English outside Brazil is gradually forming the dominant historiography for Brazil's visual arts. There is no contributor from Brazil, and the "selected bibliography," which favors exhibition catalogs, includes 11 titles specifically on the visual arts in Brazil published between the 1960s and the 2000s, 8 of them in Portuguese or Spanish and published in Latin America, among its 36 secondary sources. This proportion of titles, which corresponds to the usual prominence given to the production of constructive art from Brazil and the authors of the works listed—Aracy Amaral, Ronaldo Brito, Paulo

Herkenhoff, and a few others—are not surprising, nor is the disregard for the production of Brazilian scholars who began to publish in recent decades (Katzenstein and García 2019, pp. 216–27). Exaggerating a bit, it is as if, instead of Brazilwood, sugar, gold, coffee, or rubber, Brazil now offers works of art and primary sources for the profit of agents and institutions dominant in the global art system.

*Purity is a Myth* is another example of how scholarly production in English has surpassed academic records in Portuguese, in the case of U.S. publications. In "A History of the Field," Aleca LeBlanc presents "a history of the scholarship about a generation of avant-garde artists working primarily in Buenos Aires, São Paulo, and Rio de Janeiro in the years after the Second World War."[6] She acknowledges that the "two genealogies of the literature in English—the exhibition catalog (and) the scholarly monograph ( . . . ) never existed in Brazil or Argentina, where scholars have always contributed to varied intellectual initiatives, writing books and publishing art criticism while holding academic appointments and curating exhibitions for museums, private cultural institutions, and galleries."[7] It is worth adding that Brazil's art historiographic practices also unfolded from references other than those in English. LeBlanc also recognizes that "a new generation of scholars" emerged since the mid-1980s, in the process of democratization of those countries after dictatorships.[8] However, when focusing on the Brazilian critical production, she limits her analysis to 1970s texts by Aracy Amaral and Ronaldo Brito, naming them "Early Narratives,"[9] although from a Brazilian point of view, they could even be seen as belated, as they were written more than a decade after the artistic movements they analyze.

LeBlanc recognizes how "the passage in 1991 of the Rouanet Law (Lei 8.313), which reduces taxes for organizations that invest in cultural projects, radically changed the institutional landscape" and how "financial, energy, and communications companies began funding research, exhibitions, and publications,"[10] without including these publications in her endnotes, which mostly cites works in English published in the UK and the U.S.[11] In the first endnote, she clarifies: "I have done my best to indicate key texts in the endnotes. Any omissions are based not on the quality or importance of the research but on the constraints of space."[12] I imagine the challenge she faced in selecting the titles to quote in a few pages. Some editors of publications I contributed to in the U.S. have asked me to preferably include publications in English among the bibliographic references to offer the English-speaking audience possibilities for further reading. Given the usual size of texts, with strict word limits, balancing primordial, unavoidable, and preferable references in Portuguese, English, and other languages is seldom easy. I understand the constraints, but I would change the chapter and the book titles to "A U.S. History of the Field" and *Purity is a Myth, Decolonization Too*, respectively. Institutions that aim to carry out collaborative and effectively inclusive worldwide actions should grant more spaces for voices and references from the sociocultural contexts they intend to reach.

In this path, it is worth mentioning a quite significant passage from Adrian Anagnost's 2022 book, *Spatial Orders, Social Forms: Art and the City in Modern Brazil*: "The history of twentieth-century Brazilian art has commonly been written as a teleology culminating in 1960s artists such as Hélio Oiticica, Lygia Clark, and Lygia Pape, who are understood to have rejected the creation of discrete art objects in favor of exploring social relations. But such a story is written from the point of view of painting and sculpture and ignores the crucial, parallel history of architecture, urbanism, and city space" (Anagnost 2022, p. 16).

In this excerpt, there is no footnote indicating specific books and articles in which we can access the written "history of twentieth-century Brazilian art." The problem is not just the absence of bibliographic references but primarily what this silence signals, as if there were an unwritten consensus understood and practiced by scholars of Brazil's visual arts, which is made explicit from time to time in bibliographies such as those included in *Sur Moderno* and *Purity is a Myth*. From what I am arguing here, it would be better to say that "The history of twentieth-century Brazilian art (written in Europe and the U.S. since the 1990s and later, partially, in Brazil) has commonly been written as a teleology culminating in 1960s artists such as Hélio Oiticica, Lygia Clark, and Lygia Pape." It is essential to detach

how artists such as Alberto da Veiga Guignard, Oswaldo Goeldi, Alfredo Volpi, Iberê Camargo, and Frans Krajcberg or architects such as Vilanova Artigas and Sergio Bernardes, among others, are key figures of the Brazilian canon of art since modernism but seem not to attract much attention outside Brazil. This is unsurprising, as artistic canons vary according to whether they were crafted in more nationalist or imperialist, parochial or cosmopolitan contexts. In my view, the problem is whether the lack of reflection on this process indicates an ethnocentrism underlying the scholarly record on the visual arts from Brazil.

It is also problematic when Anagnost says this "story is written from the point of view of painting and sculpture, and ignores the crucial, parallel history of architecture, urbanism, and city space." In the history of Brazil's visual arts historiography, there have been authors—from Manuel de Araújo Porto Alegre, Ernesto da Cunha Araújo Vianna, and Mário de Andrade to Mário Pedrosa, Otilia Arantes, and Vera Beatriz Siqueira, among others—who articulate many arts (architecture, landscape design, and urbanism included) in their texts. However, as in other cultural contexts, art historians in Brazil have specialized in artistic media, periods, and regions. But this has not stopped the Brazilian canon of artistic modernity in Brazil from including painting, sculpture, architecture, landscape design, and urbanism, among other arts.

I must acknowledge that I am exploring edge cases as well as the existence of exceptions. Among many notable works, I highlight Irene Small's 2016 *Hélio Oiticica: Folding the Frame* for its thorough research of and dialogue with primary and secondary sources, and not just for that (Small 2016). Another example is *Forming Abstraction: Art and Institutions in Postwar Brazil*, in which Adele Nelson deals with the exciting mismatches and complementarities constituted by the exchanges and clashes that permeate a historiographical debate fueled in various contexts and whose "Selected bibliography" demonstrates a rare balance of dialogue with scholars inside and outside Brazil (Nelson 2022, pp. 272 (note 13), 335–38).

## 4. The U.S. (and European) Centrality

It is also necessary to consider how these historiographical practices are imbued with geopolitical tensions. One problem is subsuming Brazil into Latin America while subordinating these geopolitical units to the U.S. I do not deny that Brazil is part of Latin America, as well as the group of Portuguese-speaking countries and the network of nations constituted from the African diaspora, nor how vital this multiple geopolitical belonging is. The unequal way Latinity in the Americas is delineated from the U.S. is more difficult to accept. While all Latin American artists, groups, artistic movements, regions, and nations are lumped together in a single group, the various subfields of U.S. art are detached, creating an asymmetry that ultimately privileges the U.S. It is not a matter of nationalist exaltation since Brazil's visual arts and, in a certain way, its historiography transit between continents and cultures from the process of globalization initiated in early modernity. But it is necessary to simultaneously avoid and criticize the generalizations that have significantly affected these artistic fields.

I would agree with Eddie Chambers when he argues that "The leading scholar on Afro-Brazilian art over the past decade or so has been Kimberley Cleveland" (Chambers 2022, p. 101) if he limited his assessment to the U.S. Without this addendum, the phrase silences a discontinuous, heterogeneous, and troubled critical practice that dates back at least to 1904,[13] but which in the last quarter century gained undeniable impetus, not to mention its extraordinary momentum over the previous five years.[14] Another example of neglect (perhaps ignorance), forgetfulness, and silencing is the book *The Global Reception of Heinrich Wolfflin's Principles of Art History* (Levy and Weddigen 2020), in which three German scholars (two of whom were based in Latin America for some time) analyze the reception of that German-authored book in the Hispanic World, Mexico, and Brazil,[15] making one wonder if the book is not more about the worldwide expansion of German art historiography than its global reception. In the chapter dedicated to the reception of Wölfflin's seminal book in Brazil (Baumgarten 2020), Jens Baumgarten focuses his analysis on another German art historian, Hannah Levy, minimizing the interventions of

other agents (editors, translators, and teachers) and institutions (publishing houses and universities) in that process, in addition to not incorporating works by Brazilian authors who have been analyzing Hannah Levy's trajectory and work in Brazil (Pestana 1997; Sanajotti Nakamuta 2010; Pinheiro Machado Kern 2013, 2014, 2015).

There is no reason to keep Brazil untouched as a national entity, nor any other type of essentialism in art historiography. But the power hierarchies embedded in the global art system and the logic of the art market explain how Brazil is understood as one of Latin America's most valued regions but subordinated to the U.S. and Europe. At the same time, Germany does not disappear to make Europe stand out, nor does the U.S. in favor of Anglo-America (connecting it to anglophone Canada and the Caribbean) or North America (including Canada and Mexico). Globalization has intensified the art system's power dynamics that constitute centers and peripheries, in addition to determining graduations from tolerated to condemned nationalism.

One can argue that the problem is the language in which the worldwide art history is written, underscoring the status of English as its *lingua franca*. To understand how the situation is broader and deeper, it is worth reading Rafael Cardoso's analyses of the "structural and institutional barriers that condition how Portuguese-language authors negotiate relationships with the art-historical mainstream" (Cardoso 2019, p. 178). But when he argues that "If an interpretation is not published in English, it is probably not recognized as part of the state of the art," I would subtly revise the phrase this way: "If an interpretation is not published in English (by a publishing house based in the U.S. or the UK), it is probably not recognized as part of the state of the art." Indeed, many books and exhibition catalogs published in Brazil in the last decades include English versions of their texts.[16] However, they remain generally excluded from the debate held by publications edited in the U.S. and the UK. More than having a good command of the English language, it matters as much who writes the text as where and by whom it is published.

## 5. Brazilian Responses and Idiosyncrasies

However, it is also necessary to focus on how the worldwide artistic system transformations in recent decades have affected the art historiographical field in Brazil, with its history, dynamics, and particularities. Brazilian art historians training in Europe and the U.S. is no novelty. For decades, Brazilians have been studying abroad, conquering the "fire extinguishers"—to use Duarte's image—with which to spread the art historiographical cloud in the country.[17] More recently, Brazilian scholars began to settle and work in Europe and North America.[18] After Garcia dos Santos, we could say the North Atlantic art system has been importing South Atlantic experts to improve the training of curators, critics, and historians and, consequently, the curatorship of their collections of artworks related to Brazil and its historiography. Indeed, these Brazilian scholars can intervene and must face the challenge of changing the process focused on in this article by directly interfering in it from the North Atlantic.

Reacting to this historiographical game, scholars based in Brazil have been adopting another tactic: confronting the art historiographical debate about Brazil's visual arts in the external arena. Recognizing that its center is shifting—if it has not already moved—to the U.S., they have published in English on both sides of the North Atlantic. Sérgio B. Martins' 2013 *Constructing an avant-garde: art in Brazil, 1949–1979* (Martins 2013), Rafael Cardoso's 2021 *Modernity in Black and White* (Cardoso 2021), Renato Rodrigues da Silva's 2021 *New Perspectives on Brazilian Constructivism* (Rodrigues da Silva 2021), and Maria Berbara's 2022 *Sacrifice and Conversion in the Early Modern Atlantic World* (Berbara 2022), among other titles, are examples of academic books, not to mention a profusion of articles in peer-reviewed journals, exhibition catalogs, magazines, and websites. When announcing a newly published article on Facebook, Brazilian Art History Ph.D. candidate Gabriela Caspary commented on the historiographical problem in focus here: "I understand that our art is powerful and generates interest from international researchers. I realize that our history of art is being written here, but also outside Brazil from a foreign point of view. So,

publishing in a foreign language is relevant to disseminating our art history from here."[19] Caspary's observation indicates how researchers living in Brazil are aware of the ongoing historiographical battle, understanding the need to publish outside the country to affirm the art historiography produced there, to stay relevant in the international debate, and to try to rebalance the game.

Indeed, Art History is not a new discipline in Brazil and has a parallel history as problematic as the history of Brazil's visual arts. It can be traced back to the 1840s, based on a text by Manuel de Araújo in Porto-Alegre (Porto-Alegre 1841), which is usually seen as the inaugural landmark of a historiographical practice with moments of greater or lesser intensity, density, and singularity. Although there are not many comprehensive analyses of the art process in Brazil from time immemorial to the present,[20] nor the scrutiny of its history, this historiographical practice has been compiled[21] and critically reviewed[22] inside and outside the country. Since the mid-1980s, the historiography of Brazil's visual arts produced in the country has been gaining another scale in terms of quantity and diversity due to graduate programs and undergraduate courses, publications, and exhibitions, as the cultural context, particularly the artistic system, became more complex after the 1964–1985 civil–military dictatorship. One cannot expect to find as many academic books as in the U.S. art realm, as the Brazilian scholar evaluation system is quite different, favoring the production of articles in peer-reviewed journals, in addition to positively accepting book chapters and essays in exhibition catalogs, which determines a low incidence of single-authored academic books—not to mention the ups and downs of the publishing industry and the art system due to political and economic crises. Indeed, art criticism and historiography are not conditioned to one absolute model, much less that of the U.S.

Brazil even came to export ideas, although not precisely through scholarly records. After its curatorial revival by Paulo Herkenhoff in the 1998 São Paulo Biennial, the idea of anthropophagy as a mode of artistic and cultural relation crossed the country's borders (Herkenhoff 1998). From an artistic idea circumscribed to some moments of 20th-century Brazil's arts (literature, visual arts, music, and theatre), it was taken as a key to reading the history of art and culture in Brazil (Ferreira 2015) and even the world.[23] Thus, it became relatively dominant in Brazil's visual arts historiography inside and outside the country. One example is Caroline Jones' 2012 "Anthropophagy in São Paulo's Cold War" (Jones 2013), in which she analyzes the battle between anthropophagy and "international modernism." In that article, as Sergio Martins noted, "she touches on almost all the hot topics in the suddenly timely subject of Brazilian art—modernism, anthropophagy, the São Paulo Biennial, Concretism, Neo-concretism, Brazilian modernist architecture, Oscar Niemeyer, Lygia Clark, and Hélio Oiticica" (Martins 2014), synthesizing a complex history into trends easily inserted in the lineage of international modernism or anthropophagy, a category still somewhat linked to exoticism.

More recently, the use of anthropophagy as the connecting thread of different Brazilian artistic manifestations resounded in the catalog of *Tarsila do Amaral: Inventing Modern Art in Brazil*. Thus, there is no surprise in reading Tarsila do Amaral being connected to so many artists and art movements that are close to or distant from hers to different degrees and not necessarily directly and consciously linked to anthropophagy: Carmen Miranda, post-Neo-Concrete art, Antônio Francisco Lisboa, the *Tropicália* movement, Brazilian Baroque, José Celso Martinez Corrêa, Brazilian carnival, and Maria Bethânia (D'Alessandro and Pérez-Oramas 2017). On the other hand, also unsurprisingly, the catalog does not mention artists such as Alfredo Volpi, Rubem Valentim, and Rubem Gerchman, among others, who, whether consciously or not, dialogued with her work.

However, it is impossible to either reduce Brazil's visual arts to some media, topics, and names or to group them into a single, linear, coherent, homogeneous, and simplistically polarized narrative. In addition, artists, curators, art critics, and art historians based in Brazil and outside the country have been shaping more complex constellations.

The artistic thread that can be outlined connecting the pictorial works of Giovanni Battista Castagneto, Milton Dacosta, and Paulo Pasta is just one of the multiple "axes of

continuities and ruptures of Brazilian art in the twentieth century" in Ronaldo Brito's formulation (Brito 1983, p. 9). A set of links and disjunctions that extend for more than a century configured a historical fabric at first rarefied but singular and even potent due to how it has been reflexively but not necessarily explicitly constituted. Both external and internal references, linked to traditions engendered by poles that disseminate values in Europe, the U.S., Japan, Africa, and beyond, as well as Brazilian sociocultural peculiarities, are articulated in an unusual way that combines invention with lack of rigor, derivation with originality. Due to its innovative articulations and lapses, recurrences, and discontinuities, it is a singular, idiosyncratic, and somewhat dense history.

A history primarily engendered by artistic actions in parallel but independently of the historiography produced in that country is still quite nationalist and at the same time self-centered, oriented toward the North Atlantic, and fascinated by new trends coming from some European and U.S. centers, with which art historians in Brazil prefer to exchange and dialogue. Regarding the somewhat uncritical internalization in Brazil of external scholarly production, the Museu de Arte de São Paulo exhibition catalogs for its series of exhibitions entitled "*Histórias*"[24] (Histories) and "*Popular*"[25] (Popular) are examples of the provincial fascination for foreign authors and their scholarship. Its editors do not seem to have the same interest in dialoguing with academics from Brazilian universities as in mapping and inviting U.S.-based scholars on Brazil's visual arts to contribute texts for the referred catalogs.

The artistically engendered history of art can be seen, for example, in the dialogues that disparate artists such as Alfredo Volpi, Iberê Camargo, Lygia Pape, Antonio Dias, Cildo Meireles, Carlos Zilio, and Nuno Ramos established with the work of Oswaldo Goeldi, which constitute one of the most thought-provoking threads in the history of Brazil's visual arts. However, neither Goeldi's woodcuts and drawings—indeed, his work remains absent from the collections of the leading museums in the North Atlantic—nor this network of artistic dialogues seem to have historiographical appeal outside the country, perhaps because it does not fit the battle between anthropophagy and "international modernism." Scholars working outside Brazil are aware of these and other Brazilian artists. The issue here is not about revealing an artist lost in the jungle, not least because the art system has mapped and assimilated the visual arts of Brazil, its artists, and agents with ever greater scope and agility. The issue is the dispute over the canon and the historical narrative. In an inhomogeneous process of globalization, different artistic, historiographical, and cultural traditions still determine other valuations of artists, critics, works, ideas, and movements. The dominant threads exclude certain artists and trends and even limit the analysis of what is included; the way in which humor and libido were purged from *Lygia Pape: A Multitude of Forms* is an example of this interpretative loss.

It seems that an artistic work from Brazil will only be considered abroad if it can be entangled in the tapestry constituted with the threads and names mentioned above to introduce Brazilian art into the global art system or when they began to weave other threads, as has happened recently due to the growing interest in the arts of women, Afro-descendants, and indigenous peoples. This weaving may one day include much or everything from Brazil's arts but under the logic of the global art system, which is currently conducted from the North Atlantic, mainly from the U.S. A Brazilian point of view must not prevail, nor any nationalist bias. But to what extent will Brazil's art historiography be considered in this process?

Historiography is far from neutral but informed by its place and moment of enunciation, among other social implications. The historiography of Brazil's visual arts written outside that country is broader, more varied, and more complex than the examples I have used so far. Indeed, the extreme cases I referred to do not dominate all of Brazil's visual arts readings. Much of this scholarly record highlights aspects that the historiography produced in Brazil often does not perceive or does not face, either because of the still persistent nationalist bias or because art historians are too immersed in the historical dynamics of Brazilian art. One example is Luis Pérez-Oramas' analysis of the frames Tarsila used for her

Paris solo shows in the late 1920s, a conservative index many Brazilian critics were reluctant to face, preferring to consider them as post-cubists instead of *art deco* (D'Alessandro and Pérez-Oramas 2017, pp. 92–95).

A second example is Irene Small's analysis (Small 2017) of Ronaldo Brito's 1975 essay "Neo-concretism: Apex and Rupture of the Brazilian Constructive Project" (Brito [1975] 2017), which, since its publication, became a kind of untouchable totem in Brazilian historiography of Brazil's visual arts and had to wait more than forty years to be critically analyzed. Like Michael Asbury's (Asbury 2005, 2021) and Renato Rodrigues da Silva's (Rodrigues da Silva 2013, 2022) recent texts on Neo-Concretism, Mariola V. Alvarez's book on that artistic movement is another critical approach to this taboo topic for historians in Brazil, advancing the field. However, unfortunately, her book seems to be limited, like her dissertation (Alvarez 2012, 2023). It focuses on only parts of the movement and is another example of privilege given to a bibliography published in English in the North Atlantic over that published in Portuguese in Brazil.[26]

## 6. Keeping the Circle Open

Concluding, I focus on Ricardo Basbaum's multimedia and participatory installations (Figure 2), which he connects to the image/idea of "*terreiro de encontros*" (terrace of encounters) (Basbaum 2009, p. 202). I take it as a stimulus for a growing and better art historiographical interchange and dialogue. In Portuguese, *terreiro* means an unpaved square where people congregate for different purposes. In Brazil, *terreiro* refers particularly to coffee yards, where the beans are spread out to dry in coffee plantations, or Afro-Brazilian religious communities' congregation spaces.[27] Maria Moreira has opened a path approaching Basbaum, his trajectory, and his work to achieve fluidity in the processes of re-personalization as a strategy for the survival of Afro-Brazilian populations (Moreira 2002). Thus, it is worth exploring how his work and the image/idea of *terreiro de encontros* relate to the Afro-Brazilian universe.

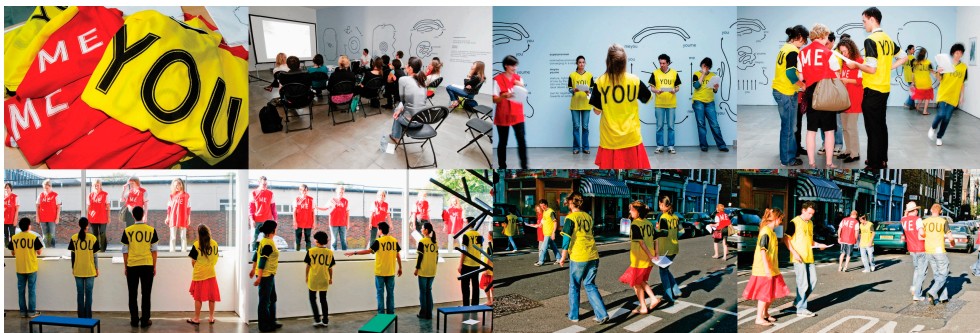

**Figure 2.** Ricardo Basbaum, *me-you: choreographies, games and exercises*, 2008, shirts, silkscreen, vinyl wall diagrams, monochrome background, performative actions, group dynamics, photography, Lisson Gallery, London. Photos courtesy of Ricardo Basbaum and the Lisson Gallery.

In this article, I connect his work to Afro-Brazilian religions' *terreiros*, particularly Tia (aunt) Ciata's house in Rio de Janeiro's "Pequena África" (Little Africa). Hilária Batista de Almeida, known as Tia Ciata, was a Bahian living in Rio de Janeiro since the late 19th century who became one of the prominent references of the Afro-Brazilian community and whose house was fundamental in the creation process of *samba* and the diffusion of other African diasporic cultural practices. Afro-Brazilian musician João da Baiana said her parties were structured with "dance in the living room, *samba* in the back of the house, and *batucada* in the yard" (Moura 1995, p. 83). Urban segregation, architectural limitations, and financial constraints did not prevent her and the Afro-Brazilian community from reinventing buildings and spaces through different daily ritualistic uses, determining an approach to architecture that is less morphological than structural. Buildings, places, objects, and people mattered less in themselves than in the relationships established in

cultural practices that ranged from the sacred to the profane, almost always mixing them. Basbaum's *terreiro de encontros* encourages me to consider the analogy between the physical and virtual environments he proposes, the ritualistic uses of Tia Ciata's house, and Michel de Certeau's conceptualization of space as a "practiced place" (de Certeau 1994, p. 202) and also to connect Basbaum's diagrams to "*pontos riscados*," Umbanda diagrams related to the Kongo graphic writing system (Martinez-Ruiz 2012) and composed of symbols that Afro-Brazilian deities usually draw on the ground at the beginning of rituals in which they manifest themselves through the bodies of the people they inhabit, erasing them before they stop manifesting in a trance. In both cases, diagrams exist for rites, although very different, subsidizing people's actions in space that aim to transform the environment and re-enchant the world, even if momentarily. Basbaum proposes his multimedia and participatory installations as *terreiros*, places for open and discontinuous artistic encounters, as Duarte offers the *Nimbo/Oxalá* circle as a space for a multidimensional exultant aesthetic experience.

Starting this article with Duarte's *Nimbo/Oxalá* and concluding it by exploring Basbaum's image/idea of *terreiro de encontros*, I follow an inclusive way of equating Africanism and art in Brazil, which considers Afro-Brazilian themes, imagery, and ideas, including authors whether of African descent or not. Abdias do Nascimento outlined it in his *Museu de Arte Negra* since the 1950s and in some 1968 texts (Do Nascimento 1980, p. 139). The idea also appears in a Clarival do Prado Valladares essay of the same year (Do Prado Valladares 1968) and Mariano Carneiro da Cunha's 1983 book chapter.[28] From 2004, Emanoel Araujo adopted and adapted this principle in the Museu Afro Brasil,[29] in São Paulo, as well as the curators of the 2018 exhibition *Histórias Afro-Atlânticas* (Afro-Atlantic Histories) held at the Museu de Arte de São Paulo and Instituto Tomie Ohtake. This inclusive understanding of the visual arts related to the African diaspora in Brazil has been discreetly preserved in the *Afro-Atlantic Histories* exhibition tour in the U.S. since 2021. As U.S. nationalism has subtly replaced the Brazilian nationalism of the original exhibition on this tour, the question remains if and how that inclusive notion of Afro-diasporic art will reverberate outside Brazil, where it has been recently questioned due to the persistent practice in the Brazilian art system of hierarchizing, devaluing, segregating, and excluding Afro-descendants and their cultural and artistic productions but also due to the reverberation of U.S. racializing modes.

Dealing with Afro-Brazilian references, Basbaum and Duarte lead me to the crossroads, home of Yoruba divinity Eshu, primordial master of communication and encounters, and Obatala's *alá*, the white mantle with which the elder orisha shelters, protects, and unites everyone. The consonances and clashes between different historiographical agents, inside and outside Brazil, have been fundamental to improving the history of its visual arts in recent decades. Basbaum proposes his installations as *terreiros*, spaces of encounters. Duarte's *Nimbo/Oxalá* is structured circularly, reviewing the *xirê*, the ritualistic circle of candomblé ceremonies, and the *rodas de samba*, samba circles, which are also spaces of encounters. Under the auspices of Eshu and Obatala, with their seemingly antithetical but complementary energies—beginning and end, youth and seniority, heat and cold, tension and calm—I envision, perhaps naively, the realm of art historiography as a *terreiro* where people can meet and, as in a *roda*, present their new texts, review old ones, and exchange ideas. Both *roda* and *terreiro* are not exempt from hierarchies and power dynamics but are situations where people can exchange, dialogue, face, re-evaluate, challenge, and overcome impasses. The crossroads can propitiate artistic and historiographic encounters, however rough or hazy they may be. Even with no illusions about the capacity of these dialogues to end hierarchies and disparities, making these historiographic tensions explicit can make this field more self-aware and dynamic. In Afro-Brazilian religious rituals, people revere Eshu at the beginning and Obatala at the end. In this article, I purposely reversed this order to keep the *roda* open and invite others to enter the *terreiro* and join the debate.

**Funding:** This research received no external funding.



**Data Availability Statement:** No additional data is available due to copyright restrictions.

**Conflicts of Interest:** The author declares no conflict of interest.

## Notes

1    On *xirê*, see (Pessoa de Barros 1999).

2    On rodas de samba, see (Moura 2004).

3    Parts of this essay were written for a 2012 book chapter that was not published in part due to some issues discussed therein and for talks given at the following events: *Mitos do Contemporâneo* (Myths of the Contemporary), organized by Rafael Cardoso and Sérgio B. Martins at Centro Cultural da Caixa in Rio de Janeiro, 2013, *New Narratives for Contemporary Art*, organized by Megan Sullivan, Christine Mehring, and Ricardo Basbaum at The University of Chicago, in Chicago, 2019, *Transnationalism and Public Space: Afro-Brazilian Art in Context*, organized by Abigail Lapin-Dardashti at The City University of New York, in New York, 2020. Besides them, I thank Claudia Mattos Avolese, Vera Beatriz Siqueira, Cecilia Fajardo-Hill, Laymert Garcia dos Santos, Alice Heeren, Aleca LeBlanc, Juliana Ribeiro da Silva Bevilacqua, Asiel Sepúlveda, and Irene Small for helpful and illuminating conversations. Thanks to Lesley A. Wolf and Gabriela Germana, editors of the Special Issue "Rethinking Contemporary Latin American Art," Sylvia Hao, and the anonymous reviewers who helped me prepare this article.

4    On this process, see (Fialho 2005a, 2014).

5    https://artandarchaeology.princeton.edu/people/irene-small (accessed on 15 December 2022).

6    Aleca LeBlanc, "A History of the Field," in (Gilbert et al. 2021, p. 263).

7    LeBlanc, "A History of the Field," p. 264.

8    LeBlanc, "A History of the Field," p. 266.

9    LeBlanc, "A History of the Field," pp. 268–70.

10    See notes 8 above.

11    LeBlanc, "A History of the Field," pp. 273–77.

12    LeBlanc, "A History of the Field," p. 273.

13    In this regard, see (Conduru 2022; Nina Rodrigues 1904; Querino 1916; Ramos 1949; Barata 1941, 1957; Do Prado Valladares 1968; Do Nascimento 1968; 1980, pp. 133–40; Zanini 1983, pp. 974–1032; Araujo 2011).

14    In addition to the texts gathered in or produced for the exhibition catalogs edited by Emanoel Araujo, which precede the creation of the Museu AfroBrasil in 2004 and, therefore, date back more than two decades, it is necessary to mention some books—(De Melo Silva and Calaça 2007; Conduru 2007; Barbosa 2020; Antonacci 2021)—and journals dossiers: Ribeiro da Silva Bevilacqua (2015); Simões (2022), among many other titles.

15    Tristan Weddigen, "Wölfflin in the Hispanic World," Peter Krieger, "Baroque and neobaroque: long-term influence of Kunstgeschichtliche Grundbergriffe in Mexico and the globalization of an idea," and Jens Baumgarten, "Wölfflin in Brazil: between Translation and Comparison," in Levy and Weddigen (2020, pp. 69–108, 109–118, and 279–288, respectively).

16    A few examples, among many others: (Amaral 1998; Roels 1998; Paraíso 2006; Andrés Ribeiro and da Silva 2008; Venosa 2008; Siqueira 2009; Souza Ayerbe and de Barros 2012).

17    Without offering an exhaustive survey of this process, it is worth remembering from Walter Zanini's graduation and doctorate at the Université de Paris VIII in 1956 and 1961, and Sonia Gomes Pereira's master's degree at the University of Pennsylvania in 1976, to the current doctorate of Renato Menezes at the École d'Hautes Études en Sciences Sociales, in Paris.

18    Ana Lucia Araujo (Howard University, Washington DC), Vivian Braga dos Santos (Institut National de l'Histoire de l'Art, Paris), Claudia Calirman (John Jay College of Criminal Justice, New York), Rafael Cardoso (Freie Universität Berlin and Universidade do Estado do Rio de Janeiro), Fernando Luiz Lara (The University of Texas at Austin), Camilla Maroja (California State University, Fullerton), Luciana Martins (Birkbeck University of London), Claudia Mattos Avolese (Tufts University, Boston), and Juliana Ribeiro da Silva Bevilacqua (Queen's University, Kingston), among others.

19    Gabriela Caspary, "Entendo que a nossa arte é poderosa e gera interesse de pesquisadores internacionais. Percebo que nossa história da arte está sendo escrita aqui, mas também fora do Brasil a partir do olhar estrangeiro. Então, publicar em língua estrangeira é relevante para a divulgação da nossa história da arte a partir daqui." 20 December 2022. https://www.facebook.com/gabicaspary (accessed on 20 December 2022).

20    Some recent examples are: (Zanini 1983), the vast and unprecedented 2000 *Brasil +500 Mostra do Redescobrimento*, curated by Nelson Aguilar, with its thirteen catalogs, the fifteen volumes of the series "Espaços da Arte Brasileira," edited by Rodrigo Naves for Cosac Naify between 1999 and 2001, and the eight-volume "Coleção Historiando a Arte Brasileira," edited by Marília Andrés Ribeiro for C/Arte between 2007 and 2019.

21    Mário Pedrosa's critical work has been collected and republished from time to time for nearly fifty years: *Mundo, Homem, Arte em Crise* (Pedrosa 1975) and *Dos Murais de Portinari aos Espaços de Brasília* (Pedrosa 1981). Edited by Aracy Amaral, *Política das Artes: Textos Escolhidos I* (Pedrosa 1995), *Forma e Percepção Estética: Textos Escolhidos II* (Pedrosa 1996), *Acadêmicos e Modernos: Textos Escolhidos III* (Pedrosa 1998), and *Modernidade Cá e Lá: Textos Escolhidos IV* (Pedrosa 2000), edited by Otilia Arantes, *Arte. Ensaios*

*Críticos—Volume I* (Pedrosa 2015b) edited by Lorenzo Mammì, and *Arquitetura* (Pedrosa 2015a). Edited by Guilherme Wisnik, and *Mário Pedrosa: Primary Documents* (Pedrosa 2016). Edited by Glória Ferreira and Paulo Herkenhoff.    Compilations of recent individual art criticism have increased: (Amaral 2006; Bittencourt 2016; Borsa Cattani 2004; Brito 2005; Chiarelli 1999; Duarte 2004; Gullar 2003; Leirner 1991; Mammì 2012; Morais 2004; Naves 2007; Osorio 2016; Pontual 2013; Roels 2010; Tavares de Araujo 2002; Venâncio Filho 2005; Wisnik 2009; Conduru 2013; Zanini 2018).    And there has been no lack of selections of texts by different authors: (Basbaum 2009), (Ferreira 2006), (Guerra 2010), (Rezende 2021).

22  In addition to critical analyzes included in the volumes cited in the previous one, other readings are: (Zielinsky 1998; Da Silva Lopes 2007; Huchet 2007; Gonçalves Terra 2010; Boudon-Machuel 2013; Menezes 2018).

23  Two examples: (Cocco and Cava 2018; Refskou et al. 2019).

24  "*Histórias da Infância*" (Histories of Childhood) in 2016, "*Histórias da Sexualidade*" (Histories of Sexuality) in 2017, "*Histórias Afro-Atlânticas*" (Afro-Atlantic Histories) in 2018, "*Histórias das Mulheres: Artistas até 1900*" (Histories of Women: Artists before 1900) and "*Histórias Feministas: Artistas depois de 2000*" (Feminist Histories: Artists after 2000) in 2019, "*Histórias da Dança*" (Histories of Dance) in 2020, and "*Histórias Brasileiras*" (Brazilian Histories) in 2022.

25  "*Portinari Popular*," "*Tarsila Popular*," and "*Volpi Popular*" were presented in 2016, 2019 and 2022, respectively.

26  The comparison between names and works cited in the "Bibliography," "Notes," and "Index" indicates what she highlights in the existing bibliography, what she suggests as further reading, and with whom she dialogues.

27  On *terreiros*, see (Sodré 1988).

28  Marianno Carneiro da Cunha, "Arte Afro-Brasileira".

29  Renamed Museu Afro Brasil Emanoel Araujo in 2022.

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
