# Peer review of "Roda and Terreiro: The Historiography of Brazil’s Visual Arts at the Crossroads of Globalization"

_arts, 2022_

Round 1

Reviewer 1 Report

Analyzing the historiography of Brazilian visual arts in the frame of globalization is quite relevant as it contributes to rewriting and reviewing the canonical narratives of Art History from the Brazilian point of view. It is an opportunity to decenter the main speech and the canon. The author points out the predominance of foreign historiography in a critical way.

Besides its importance, the article needs deep reformulation in terms of its structure. It should clearly identify the research question, address it in a broad context and highlight the purpose of the study – the need for a revision of the Brazilian historiography –, including the article's contribution to the topic, and the main methods applied also should be mentioned. Still, it should explain the interpretation of the results, how the narratives take their flow, and the selection of authors analyzed and why.

There are no title sections, neither an introduction, nor a conclusion section clearly identified. The article has a weak scientific structure.

Some detailed comments:

Section 1) starts mentioning Duarte’s work as a metaphor for Art History’s globalization but does not give any context to readers about it nor the purposes of the metaphor. The questions raised at this point are not clear (lines 21-28).

The first section presents an updated bibliography of foreign contributors to the historiography of Brazilian visual arts, particularly from the first decade of the XXI century, and most probably due to the economic growth of Brazil during the period and the attention that has been caught from international academic scholars.

In sections 5) and 6) the proposed cases of Afro-Brazilian references – terreiro dos encontros – should be better contextualized and developed, according to the idea of diasporic cultural practices, and in comparison, with the achievements of the revised historiography as spaces of encounters.

Throughout the article, some concepts are referred to but never defined by the author. The reader should understand the author’s perspective on, for example, globalization, modernism, imperialism, (self-)colonialist, nationalism (line 17, 136), anti-colonial, de-centered (line 27), peripheries (line 251), anthropophagy (line 315), ethnocentrism (line 362).

The article has a corpus of updated bibliography in terms of exhibitions and artists with international recognition, but not so relevant in terms of the art system although it mentions it (line 124, for example). I would recommend the reading of Ana Letícia Fialho and Maria Amélia Bulhões, both authors have several contributions to the topic.

Author Response

Thanks very much for your review. It helped me a lot.

I reformulated the article, re-arranging the sessions, cutting certain parts and moving others, including a new introduction and reviewing the conclusion–though not naming them as introduction and conclusion–, besides adding titles for each section.

Best regards

I also tried to clarify certain terms and concepts.

I tried to better present Duarte's and Basbaum's works and Afro-Brazilian cultural references.

I know the work of Ana Letícia Fialho and Maria Amélia Bulhões, and opted to dialogue with Fialho.

Reviewer 2 Report

The premise of the article is very true and I found it interesting, especially because I am one of those American scholars writing about Brazil who has often found it impossible to use references from Brazil and feel bad about that.  Too little written for the "global" audience except in Portuguese - this is a limitation.  While it is true, we are "outsiders" it is up to Brazilian scholars and writers to correct that perception by contributing.  I was also wondering about the African influences you speak about in abstract, but do not really follow through with in much detail.   I was looking forward to that.  

Author Response

Thanks very much for your review. It helped me a lot.

I reformulated the article, re-arranging the sessions, cutting certain parts and moving others, including a new introduction and reviewing the conclusion–though not naming them as introduction and conclusion–, besides adding titles for each section.

I also tried to clarify certain terms and concepts.

I tried to better present Duarte's and Basbaum's works and Afro-Brazilian cultural references.

Best regards

Reviewer 3 Report

This is an interesting and thorough survey of the state of the research in the field and it's US bias. I was intrigued by the opening and closing paragraphs referring to Duarte's Nimbo/Oxala, but felt that you needed to say more about this project and how exactly you see it as relating to your core argument. Other aspects that it woudl be worth slightly developing relate to the core literature on the 'global turn' (Aruna da Souza, Hans Belting etc.) in art history and issues with this, on the one hand, and 'decoloniality' on the other hand (e.g. Walter Mignolo) other. I think a clearer statement about the centrality of African cultural experience to Brazilian culture could come sooner and be more of a focus. More could usefully be said about the focus of international scholars on Rio and SP and reasons for this, and about the institutional entanglements between Brazilian art institutions and US politically-driven philanthropic funding. These issues need not be explored in depth as your focus lies elsewhere, but they would held to lend further depth to your arguments.    

Author Response

Thanks very much for your review. It helped me a lot.

I reformulated the article, re-arranging the sessions, cutting certain parts and moving others, including a new introduction and reviewing the conclusion–though not naming them as introduction and conclusion–, besides adding titles for each section.

I also tried to clarify certain terms and concepts.

I tried to better present Duarte's and Basbaum's works and Afro-Brazilian cultural references.

I opted to dialogue with Aruna D'Souza, and Piotr Piotrowski regarding global art history, and with Anibal Quijano concerning coloniality.
Best regards

Round 2

Reviewer 1 Report

The a

The authors have indeed positively reformulated the article, introducing sections and new bibliography entries. The article is placed in a theoretical broad context and highlights its importance.

However, it is still missing a clear definition of the purpose of the work in the introduction, and I think for readers would be an advantage if the authors clarify that the methodology in use is a theoretical one.

Author Response

Thank you again for your readings and comments.

I tried to incorporate your suggestions to solve the problem – "it is still missing a clear definition of the purpose of the work in the introduction, and I think for readers would be an advantage if the authors clarify that the methodology in use is a theoretical one" – with the following review in the third paragraph: "Using a theoretical methodology, delineating a brief panorama of the recent historiographic production on Brazil's visual arts, especially those produced outside that country, dialoguing with authors who previously focused on this topic, and analyzing artworks, exhibitions, institutions, and, above all, crucial passages from some publications (focusing mainly on academic books), this article makes this discussion more explicit, highlights some hot spots and calls other authors to engage in the debate."

I also made other additions and editions to the text, which you can read in the attached file.
